# C9ORF72 GGGGCC repeat-associated non-AUG translation is upregulated by stress through eIF2α phosphorylation

Weiwei Cheng[1], Shaopeng Wang[1], Alexander A. Mestre[1], Chenglai Fu [2], Andres Makarem[1], Fengfan Xian[1], Lindsey R. Hayes[3], Rodrigo Lopez-Gonzalez[4], Kevin Drenner[5], Jie Jiang[5], Don W. Cleveland[5] & Shuying Sun[1]

Hexanucleotide repeat expansion in *C9ORF72* is the most frequent cause of both amyotrophic lateral sclerosis (ALS) and frontotemporal dementia (FTD). Here we demonstrate that the repeat-associated non-AUG (RAN) translation of $(GGGGCC)_n$-containing RNAs into poly-dipeptides can initiate in vivo without a 5′-cap. The primary RNA substrate for RAN translation of C9ORF72 sense repeats is shown to be the spliced first intron, following its excision from the initial pre-mRNA and transport to the cytoplasm. Cap-independent RAN translation is shown to be upregulated by various stress stimuli through phosphorylation of the α subunit of eukaryotic initiation factor-2 (eIF2α), the core event of an integrated stress response (ISR). Compounds inhibiting phospho-eIF2α-signaling pathways are shown to suppress RAN translation. Since the poly-dipeptides can themselves induce stress, these findings support a feedforward loop with initial repeat-mediated toxicity enhancing RAN translation and subsequent production of additional poly-dipeptides through ISR, thereby promoting progressive disease.

[1] Department of Pathology and Brain Science Institute, Johns Hopkins University School of Medicine, Baltimore, MD 21205, USA. [2] The Solomon H. Snyder Department of Neuroscience, Johns Hopkins University School of Medicine, Baltimore, MD 21205, USA. [3] Brain Science Institute and Department of Neurology, Johns Hopkins University School of Medicine, Baltimore, MD 21205, USA. [4] Department of Neurology, University of Massachusetts Medical School, Worcester, MA 01605, USA. [5] Ludwig Institute for Cancer Research and Department of Cellular and Molecular Medicine, University of California at San Diego, La Jolla, CA 92093, USA. Correspondence and requests for materials should be addressed to S.S. (email: shuying.sun@jhmi.edu)

C9ORF72 hexanucleotide repeat expansion is the most common cause of sporadic and familial amyotrophic lateral sclerosis (ALS) and frontotemporal dementia (FTD)[1, 2]. This mutation connects ALS/FTD to a heterogeneous class of repeat expansion-associated neurological diseases[3, 4]. RNA-mediated toxicity is believed to be one major disease-causing mechanism when the repeats are located in non-coding regions[5–7].

There are two alternative but not mutually exclusive hypotheses for gain-of-toxicity from the RNA isoform of *C9ORF72* with the repeat expansion in the first intron[8, 9]. First, RNA foci formed by hexanucleotide repeats that could sequester RNA-binding proteins (RBPs) and disrupt RNA processing. Second, toxicity may derive from aberrant accumulation of dipeptide repeat

(DPR) proteins produced by repeat-associated non-AUG (RAN) translation in all reading frames. Both possibilities may converge on dysfunction of nucleocytoplasmic transport as a driver of disease pathogenesis[10–13]. Aggregation of all five DPRs (poly-GA, poly-GR, poly-PA, poly-PR, and poly-GP) translated from both sense GGGGCC[14–18] and antisense CCCCGG[14, 19–21] repeat-containing RNAs have been identified in C9ORF72 patients. When overexpressed by AUG-driven translation of poly-dipeptides but using degenerate codons so as to eliminate the RNA repeats in yeast[10], cultured cells[22–30], fly[23, 24, 31, 32], and mouse[12, 33], several of the DPR proteins generate various toxic effects. Therefore, an approach to decrease the levels of these toxic polydipeptides by inhibiting RAN translation holds great therapeutic promise. Moreover, methods to reduce DPRs without

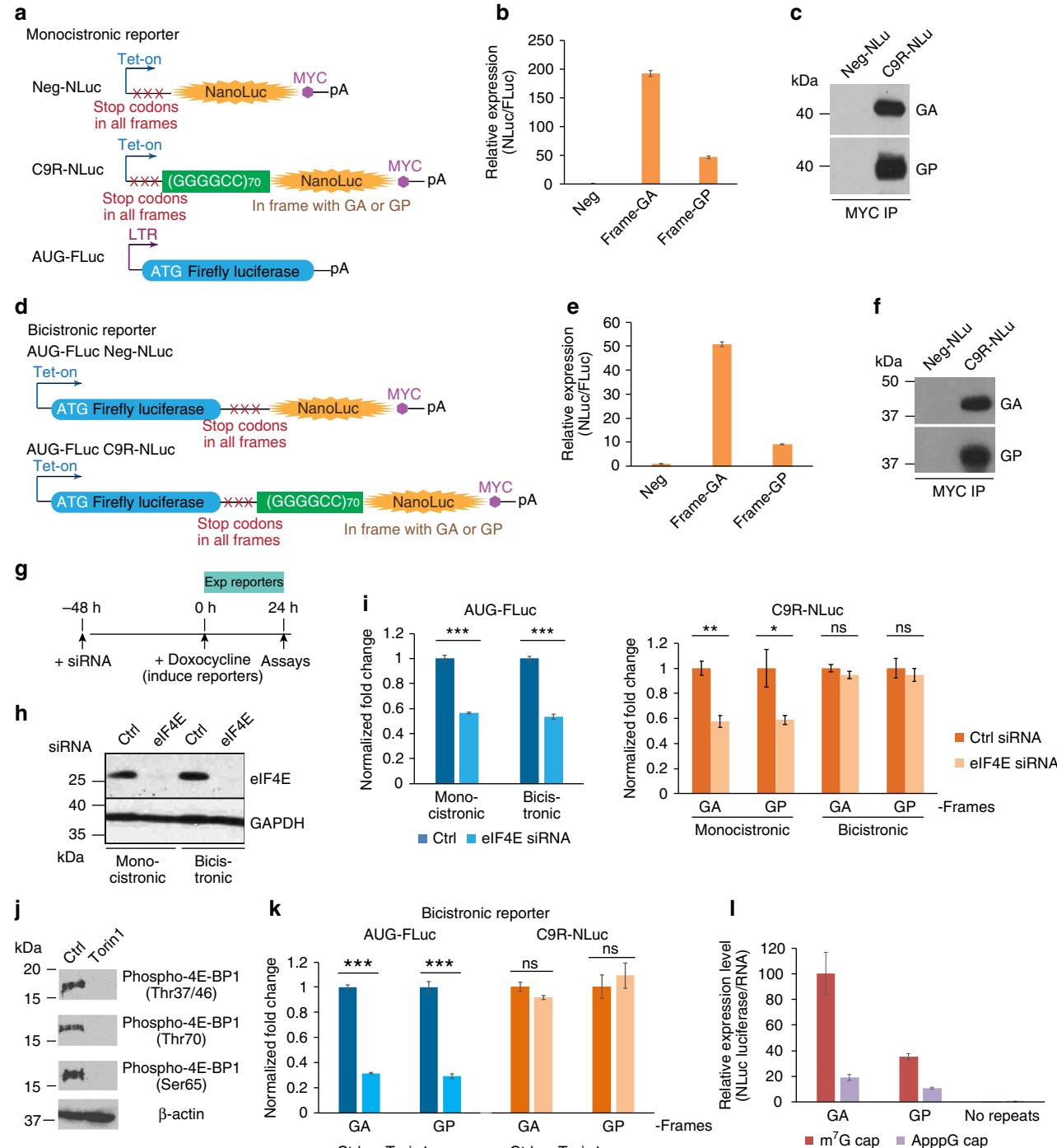

changing accumulation of the RNA repeats may enable dissecting the pathogenic pathways mediated by RNA vs. dipeptides.

The translation of most eukaryotic mRNAs involves recognition of a 5'7-methylguanosine (m⁷G) cap, formation of a pre-initiation complex, scanning on the mRNA to an AUG start codon, and assembly of the 80S ribosome to initiate translation[34]. RAN translation bypasses the requirement for an initiating AUG and has been found in several repeat expansion disorders, including myotonic dystrophy (CUG and CAG repeats[35]; CCUG and CAGG repeats[36]), spinocerebellar ataxia type 8 (CAG repeats)[35], Fragile X-associated tremor/ataxia syndrome (CGG[37] and CCG repeats[38]), and Huntington's disease (CAG repeats)[39]. Previous work on CGG repeats located in the 5' untranslated region (UTR) of *FMR1* reported that RAN translation requires cap-dependent ribosomal scanning in vitro[40]. However, one unique feature of the *C9ORF72* expansion is that the repeat is located in an intron, which is normally excised intranuclearly into a lariat structure, debranched and degraded by exonucleases[41]. Correspondingly, after splicing, all intron-derived RNAs will not have the co-transcriptionally added 5'-m⁷G cap that is required for typical translation initiation. How then does RAN translation occur?

Many viral RNAs and a handful of cellular RNAs can start translation in a cap-independent manner by bypassing the requirement of some of the initiation components, utilizing instead an internal ribosome entry site (IRES)-mediated pathway[42, 43]. IRESes are usually complex RNA structures that directly recruit certain translation initiation factors to the internal sites within RNA transcripts and have been proposed to direct ribosome assembly without RNA scanning[42, 43]. Such translation may act as a "fail-safe" mechanism to maintain or promote translation of selected cellular RNAs under stress conditions when cap-dependent translation is downregulated, thereby restoring cellular homeostasis through what is known as the "integrated stress response" (ISR)[44]. Previous work has demonstrated that RAN translation is strongly influenced by the secondary structure of the repeat RNA[35, 39]. Whether the translation of *C9ORF72* expanded intronic repeats is analogous to this type of translation and how RAN translation responds to stress has not been established. This is of particular relevance to ALS/FTD as stress responses and stress granule alteration have been increasingly associated with adult-onset progressive neurodegenerative diseases[45].

Here we demonstrate that a GGGGCC repeat-containing spliced intron is exported to the cytoplasm and serves as the main RNA template for *C9ORF72* sense repeat translation. This translation is shown to be 5'-cap-independent, but with an

initiation efficiency lower than the cap-dependent translation. Cap-independent RAN translation is shown to be upregulated by various stress stimuli that drive phosphorylation of the α subunit of eukaryotic initiation factor-2 (eIF2α), the core event of ISR. Further, expression of the TDP-43 prion-like domain promotes stress granule formation, elevates eIF2α phosphorylation, and enhances RAN translation. The stress-induced RAN translation upregulation can be reduced by small molecule compounds inhibiting the phospho-eIF2α pathway. Our results identify how translation initiation is triggered by expanded repeat-containing RNAs and establish that one or more initial stresses arising from repeat-mediated toxicity may trigger a feedforward loop to generate more and more toxic DPRs that contribute to irreversible neurodegeneration.

## Results

**(GGGGCC)ₙ RAN translation can initiate without 5'-cap.** In order to understand the *C9ORF72* repeat-associated translation initiation in vivo, we developed a series of stable cell lines expressing dual-luciferase reporters, one whose encoded protein can be produced only through RAN translation (Nanoluc Luciferase or NLuc) and one whose product is generated by AUG- and cap-dependent canonical translation (Firefly Luciferase or FLuc). To monitor RAN translation efficiency in a timely manner in vivo, we constructed tetracycline-inducible reporters and engineered each reporter in a single genomic locus in HeLa Flp-In cells using a site-directed recombinase (Flip)[46]. We fused around (GGGGCC)₇₀ repeats with NLuc lacking an AUG start codon and containing a C-terminal MYC tag (C9R-NLuc) in-frame with either poly-GA or poly-GP (Figs. 1a, d). We also included 99nt of the intronic sequences before the expanded repeats in the *C9ORF72* pre-mRNA to maintain the context for RAN translation of *C9ORF72* repeats. Multiple stop codons in all three reading frames were included 5' to the repeats to prevent any leakage from canonical translation. We used NLuc without an AUG or the (GGGGCC)₇₀ repeats (Neg-NLuc) as negative controls to define the assay background (Figs. 1a, d), and an FLuc reporter with an AUG start codon (AUG-FLuc) as a positive internal control (Figs. 1a, d).

We first generated monocistronic reporter genes to encode RAN translation or canonical translation from separate RNA transcripts. We integrated C9R-NLuc into the inducible Flp-In reporter site and used retroviral integration for AUG-FLuc to generate stable cell lines expressing both (Fig. 1a). In this system, both transcripts will have the m⁷G caps and poly(A) tails. The C9R-NLuc in both GA and GP reading frames showed much

**Fig. 1** RAN translation of *C9ORF72* hexanucleotide repeats can initiate with and without 5'-cap. **a** Schematic of monicistronic dual-luciferase reporters for RAN translation and canonical AUG translation. **b** HeLa Flp-In cells were induced to express translation reporters by doxycycline for 24 h. Relative RAN translation products from Frame-GA and Frame-GP were compared to no-repeat control. NLuc signals were normalized to FLuc in each sample. **c** Immunoprecipitation using MYC antibody from cells expressing Neg-NLuc or C9R-NLuc, followed by immunoblotting with GA or GP antibody. **d** Schematic of bicistronic reporters for cap-independent RAN translation and AUG translation. **e** Relative RAN translation products from Frame-GA and Frame-GP were compared to no-repeat control. NLuc signals were normalized to FLuc in each sample. **f** Immunoprecipitation using MYC antibody from bicistronic reporter cells followed by immunoblotting with GA or GP antibody. **g** Schematic timeline of siRNA transfection and induction of luciferase reporters in HeLa Flp-In cells. **h** Reporter cells were transfected with non-targeting siRNA or siRNA against cap-binding protein eIF4E. Immunoblotting of eIF4E showed the knockdown efficiency. GAPDH was blotted as internal control. **i** Expression of AUG-FLuc translation (left) and C9R-NLuc RAN translation (right) reporters in presence of eIF4E siRNA compared to non-targeting siRNA control. *P < 0.05, **P < 0.005, ***P < 10⁻⁶, two-tailed t-test. **j** Reporter cells were treated with mTOR inhibitor Torin 1 for 24 h. Immunoblotting of phospho-4E-BP1 using antibodies recognizing different phosphorylation sites. β-actin was blotted as internal control. **k** Expression changes of AUG-FLuc and C9R-NLuc in bicistronic reporter cells under mTOR pathway inhibition by Torin 1 treatment. **P < 0.005, ***P < 0.0005, two-tailed t-test. **l** The relative translation levels of C9R-NLuc (Frame-GA and Frame-GP) with and without a functional 5'-cap. HeLa cells were transfected with in vitro transcribed monocistronic C9R-NLuc and Neg-NLuc RNA with either 5'-m⁷G cap or ApppG cap analog. The NLuc luciferase was normalized to RNA level in each condition, and each sample was compared with 5'-m⁷G capped C9R-NLuc in frame-GA (set as 100). Data are mean ± SEM from three biological replicates

higher levels of activity than Neg-NLuc, confirming non-AUG translation driven by the $(GGGGCC)_{70}$ repeats (Fig. 1b). Immunoprecipitation for the MYC epitope encoded in either GA or GP frames followed by immunoblotting with GA or GP antibodies revealed products corresponding to the expected size of $GA_{70}$-NLuc or $GP_{70}$-NLuc (Fig. 1c), outcomes suggesting a translation initiation site close to the beginning of the repeats.

Next, we used a bicistronic reporter to test whether the RAN translation of C9 repeats can occur independent of a 5′-cap. In this approach, the first cistron (AUG-FLuc) is translated from the 5′ end by canonical cap- and AUG-dependent initiation and is terminated by stop codons placed in all reading frames. This is followed by the second ORF encoding C9R-NLuc, which is translated only if the repeats can recruit ribosomes by a cap-independent mechanism (Fig. 1d). Comparing to Neg-NLuc, the C9R-NLuc again showed significantly higher expression levels, indicating that the RAN translation of C9 repeats can initiate without 5′-cap (Fig. 1e), although at an efficiency lower than the cap-dependent version. Immunoprecipitation/immunoblotting revealed similar translation products as were produced from the monocistronic reporters (Fig. 1f).

To test that NLuc and FLuc are indeed encoded from the same RNA transcript, rather than NLuc encoded by a separate monocistronic RNA (e.g., produced by a cryptic promoter), we transfected reporter cells with an siRNA targeting FLuc-coding sequences. The FLuc siRNA efficiently reduced luciferase activity and RNA levels of both FLuc and NLuc (Supplementary Fig. 1a,b). It is noted that reduction of Fluc luciferase activity was slightly more than the reduction in RNA, possibly due to both RNA degradation and translational repression caused by the FLuc-targeting siRNA. In any case, the reduction of NLuc luciferase activity (and the RNA encoding it) was comparable to that for FLuc (Supplementary Fig. 1a, b), demonstrating the two luciferases are translated from one RNA transcript.

To further confirm the cap-independent translation of C9R-NLuc in the bicistronic reporter, we induced reporter expression after siRNA knockdown of the eukaryotic translation initiation factor 4E (eIF4E; Figs. 1g, h), the cap-binding protein that facilitates initiation complex assembly and mRNA scanning[34]. As expected, reduction of eIF4E decreased AUG-FLuc translation in both systems and the C9R-NLuc in the monocistronic reporter (Fig. 1i). However, there was no reduction of C9R-NLuc in the bicistronic reporter (Fig. 1i), indicating that the 5′-cap and the cap-binding protein eIF4E are not essential for the translation initiation of $(GGGGCC)_{70}$ repeats in vivo.

In addition, the cap-dependent translation initiation is stimulated by cytokines or growth factors for cell growth and proliferation through the mammalian target of rapamycin (mTOR) signaling pathway via the eIF4E-binding protein (4E-BP)[47]. 4E-BP1 binds to eIF4E and prevents recruitment of the translation machinery to mRNA. Activation of the mTOR pathway induces hyperphosphorylation of 4E-BP1, disrupts its eIF4E-binding activity, and therefore enhances the cap-dependent translation[47]. We therefore used mTOR inhibitor as an alternative approach to inhibit cap-dependent translation and tested how RAN translation is affected. We treated cells with Torin 1, an inhibitor directly targeting the mTOR catalytic site and inhibiting phosphorylation of both mTORC1 and mTORC2 substrates[47]. As expected, treatment with Torin 1 reduced 4E-BP1 phosphorylation (Fig. 1j) and decreased the AUG-FLuc translation and the C9R-NLuc in the monocistronic reporter (Supplementary Fig. 1c), but there was no reduction of C9R-NLuc in the bicistronic reporter (Fig. 1k), highly consistent with the eIF4E knockdown results (Fig. 1i).

We further examined the cap-independent $(GGGGCC)_{70}$ translation and quantitatively compared the relative translation initiation efficiency with and without 5′-cap using in vitro synthesized RNA. We generated the monocistronic C9R-NLuc and Neg-NLuc RNA by in vitro transcription containing either the normal 5′-m⁷G or the non-functional cap analog ApppG, and transfected into cells. We observed that C9R-NLuc RNA produced significantly higher amount of protein (luciferase activity) than Neg-NLuc RNA, even without a functional cap (Fig. 1l), confirming the capability of cap-independent translation from $(GGGGCC)_{70}$ repeats. Cellular IRES-mediated translation is typically less efficient than cap-dependent translation[42]. The comparison in our test cells revealed that cap-independent repeat RNA translation initiation is about 20–30% efficiency of the cap-dependent translation (Fig. 1l).

**Cap-independent RAN translation is upregulated upon stress.** We next tested whether stress affected RAN translation of the $(GGGGCC)_{70}$ repeats using either low-dosage sodium arsenite-induced oxidative stress or MG132-induced unfolded protein stress[48]. Six hours after addition of either stressor and induction (with doxycycline) of the reporter genes, the levels of newly synthesized FLuc and NLuc luciferases were measured (Fig. 2a). The relative NLuc and FLuc RNA levels were constant with and without stress when encoded by the bicistronic genes or when expressed from monocistronic reporter genes (Supplementary Fig. 2a). Therefore, any differences in the luciferase activities directly represent differences in translation efficiency. As expected[44], AUG-FLuc expression from canonical translation was reduced under stress conditions (Fig. 2b). C9R-NLuc translation was reduced modestly in monocistronic reporters (Supplementary Fig. 2b), but markedly upregulated in the bicistronic reporters (Fig. 2b). As the elevation was not observed in the cap-dependent reporter, it is unlikely due to the protein stability changes, but rather to be caused by stress-induced upregulation of cap-independent RAN translation. In addition, mirroring RAN translation elevation, both arsenite and MG132 induced formation of stress granules (Fig. 2c), compartments for storage of stalled translation machinery under stress[45], and activated eIF2α phosphorylation in reporter cells (Fig. 2d).

**RAN translation is upregulated by phospho-eIF2α upon stress.** One key component mediating cellular stress with translation regulation is the translation initiation factor eIF2α. During ISR, diverse stress signals converge at a single phosphorylation event on serine 51 of eIF2α. This can lead to global protein synthesis reduction, paradoxically coupled with elevated translation of a subset of mRNAs (Fig. 3a). This includes mRNAs containing short upstream open reading frames (uORFs) in the 5′ UTRs with translation re-initiation at downstream coding sequences, and IRES-containing mRNAs with cap-independent translation[44]. We therefore tested whether the stress-induced RAN translation elevation is mediated through eIF2α phosphorylation.

We first examined whether treatment of cells with small molecule inhibitors of the eIF2α pathway can block stress-induced translation changes. ISRIB is a recently identified compound that is believed to block downstream signaling of phospho-eIF2α without changing the level of eIF2α phosphorylation[49] (Fig. 3a). GSK2606414 (PERKi) inhibits PRKR-like ER kinase (PERK)[50], one of the kinases activated by unfolded protein response (UPR) and phosphorylating eIF2α[51] (Fig. 3a). Treatment with either of these two inhibitors (Fig. 3b) markedly inhibited arsenite- and MG132-induced stress granule formation (Figs. 3c, d). Translation alterations induced by either arsenite or MG132 were substantially reversed (Fig. 3e), indicating that phospho-eIF2α is indeed involved in stress-induced RAN translation elevation.

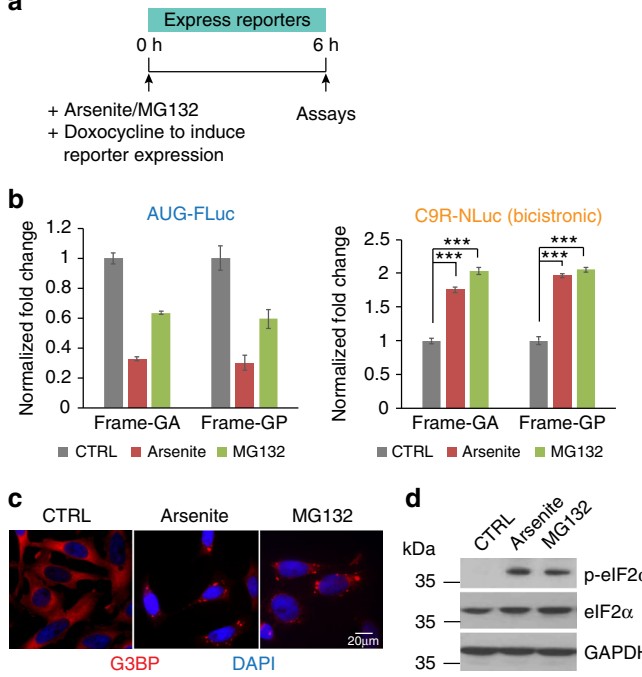

**Fig. 2** Cap-independent RAN translation is upregulated upon stress. **a** Schematic timeline of stress stimuli and induction of luciferase reporters in HeLa Flp-In cells. **b** Expression of bicistronic luciferase reporters was induced by doxycycline for 6 h in HeLa Flip-In cells. At the same time, cells were treated with arsenite or MG132 stimuli. Relative expression of AUG-FLuc and C9R-NLuc reporters was compared with no stress control. The luciferase signals were normalized to the total protein amount. Data are mean ± SEM from three biological replicates. ***$P < 0.0005$, two-tailed $t$-test. **c** Stress granules were detected with G3BP immunofluorescence in reporter cells treated with arsenite and MG132. **d** Immunoblotting using antibody recognizing phospho-eIF2α at the Ser51 site showed the upregulation under stress. GAPDH and total eIF2α were blotted as internal control

To test whether the mTOR signaling pathway is involved in the stress-induced translation changes, we examined the phosphorylation status of multiple sites of 4E-BP1 by immunoblotting. We observed that phosphorylation of Thr37/46, but not Thr70 or Ser65, decreased upon arsenite treatment (Supplementary Fig. 2c). As it has been shown that the combination of phosphorylation events at all these sites is essential to change the 4E-BP1 association (with eIF4E[52] and Thr37/46 alone having no effect on translation[53]), it is unlikely that the stress-induced RAN translation and AUG translation changes in our assay are due to the mTOR signaling pathway.

To test whether eIF2α phosphorylation is sufficient to promote RAN translation, we also expressed either wild-type or phospho-mimetic mutant S51D (serine 51 to aspartic acid) of eIF2α at modest levels in our reporter cell lines (Fig. 3f, supplementary Fig. 2d). Wild-type eIF2α enhanced reporter translation in general, as expected (Supplementary Fig. 2e). The S51D mutant had minimum effect on AUG-FLuc and on the monocistronic C9R-NLuc expression, but specifically promoted C9R-NLuc expression in the bicistronic reporter (Fig. 3g, Supplementary Fig. 2e). As the relative RNA levels were not altered (Supplementary Fig. 2f), these findings demonstrate that the phospho-mimetic S51D mutant of eIF2α enhances cap-independent RAN translation. Furthermore, treatment with ISRIB inhibited S51D-stimulated RAN translation (Fig. 3h). Collectively, these data

demonstrate that eIF2α phosphorylation is the key component regulating RAN translation under ISR.

In addition, we also compared the stress response of RAN translation with the IRES translation of cellular RNAs. Activating transcription factor 4 (ATF4) is an important genetic regulator of the UPR and it has been well established that translation initiation of ATF4 mRNA is activated by stress-induced eIF2a phosphorylation[54]. In the canonical isoform, this is mediated by re-initiation at coding regions downstream of two uORFs in the 5′ UTR[55]. An alternatively spliced isoform contains a highly structured region in the 5′UTR, which is responsible for internal ribosome entry of cap-independent translation and stress-induced upregulation[56]. Correspondingly, after replacing the C9 repeats in our bicistronic reporter with the 5′UTR of the longer ATF4 mRNA isoform, we observed that translation of NLuc was still upregulated by arsenite addition or eIF2α S51D expression, with this elevation suppressed by ISRIB treatment (Supplementary Fig. 2g). Collectively, these data indicate that C9 repeat-driven RAN translation and IRES translation are elevated by stress through similar mechanisms.

**TDP-43 prion-like domain enhances RAN translation through ISR.** Cytoplasmic mis-localization and inclusion of TDP-43 are widely present as a pathological feature in almost all ALS and half of FTD patients, including disease caused by hexanucleotide expansion in *C9ORF72*[9]. Dysfunction of TDP-43 has been shown to induce stress granule formation and eIF2α phosphorylation[57]. The prion-like domain at the C-terminal fraction of TDP-43 is believed to underlie pathogenesis[58]. We therefore tested whether expression of this domain influences C9 RAN translation. We expressed either wild-type TDP-43 or its C-terminal prion-like domain (TDP43-F4) fused with EGFP[59] in our reporter cells. Consistent with previous findings, wild-type TDP-43 was predominantly nuclear, while TDP43-F4 showed cytosolic localization and inclusions (Fig. 4a). TDP43-F4, but not wild-type TDP-43, induced stress granules (Figs. 4a, b), accompanied by accumulation of phopho-eIF2α (Fig. 4c). TDP43-F4 also promoted cap-independent C9R-NLuc translation in our bicistronic reporter (Fig. 4d), without affecting the corresponding RNA level (Supplementary Fig. 2h). Treatment with ISRIB and PERKi compounds inhibited stress granule induction (Fig. 4a, b) and translation alterations (Fig. 4f), accompanied by reduced phospho-eIF2α and (when PERKi was added) phospho-PERK (Fig. 4e). These findings show that TDP-43 dysfunction can activate ISR pathways and thereafter enhance RAN translation of C9ORF72 expanded repeats to produce more DPRs.

**RAN translation of GGGGCC repeats in *C9ORF72* spliced intron.** We next constructed bicistronic, doxycycline-inducible reporters in which the $(GGGGCC)_{70}$ repeats were located within the *C9ORF72* first intron. The *C9ORF72* exons 1a and 2, as well as ~200 bases of intronic sequences adjacent to the 5′ and 3′ splice sites were included, and the NLuc-coding sequence was inserted in the intron 3′ to the $(GGGGCC)_{70}$ repeats in frame with either poly-GA or poly-GP. Finally, the FLuc-coding sequences were fused to exon 2 in frame with the *C9ORF72* AUG start codon (Fig. 5a), and the final construct was integrated into the unique Flp-In site in our HeLa cells. We verified that the reporter pre-mRNAs were correctly spliced with an efficiency that was not affected by the presence of the expanded repeats (Supplementary Fig. 3a). RAN translation was again observed, with a much higher C9R-NLuc activity than the no repeat control (Fig. 5b). Immunoprecipitation with MYC antibody confirmed the expression of GA-NLuc or GP-NLuc proteins (Fig. 5c), similar to what was seen with our previous reporter systems (Fig. 1c, f). Relative RAN

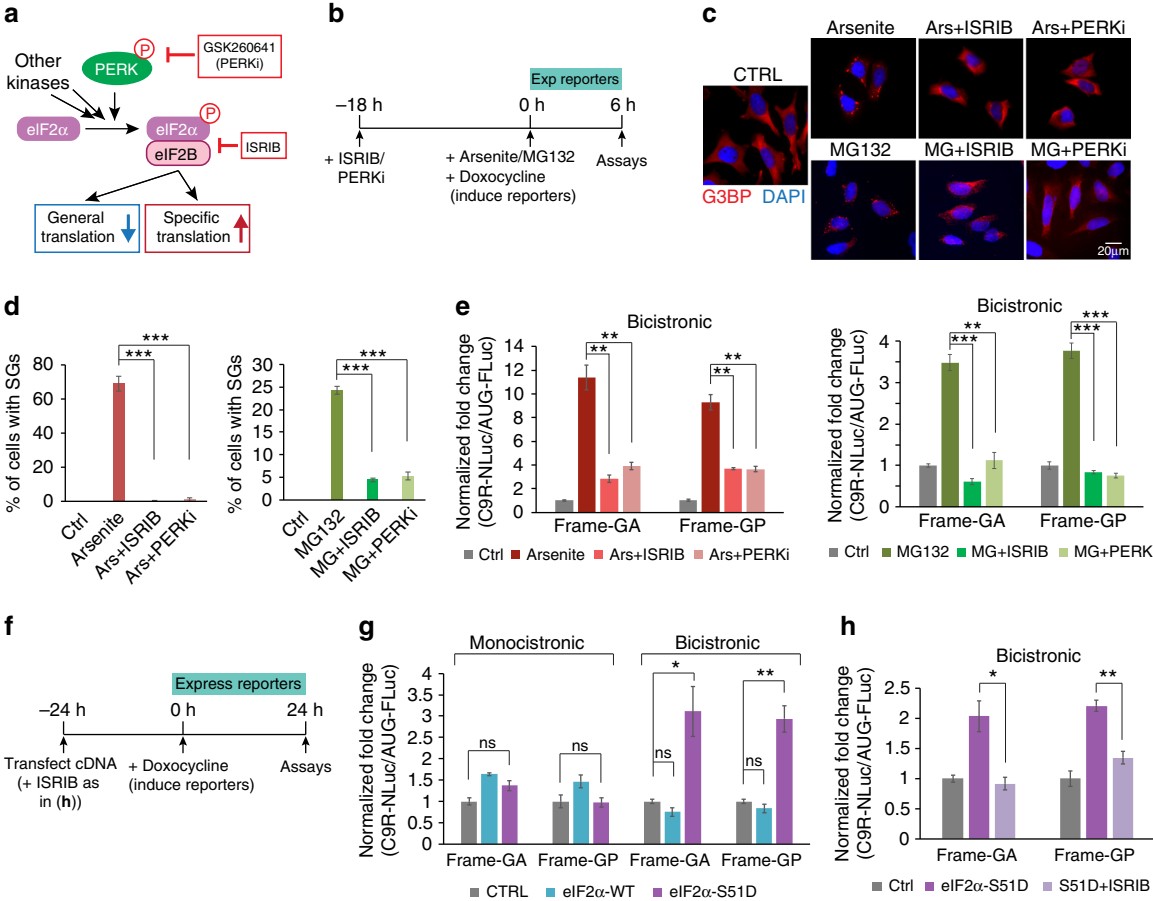

**Fig. 3** Stress-induced RAN translation elevation is mediated through eIF2α phosphorylation. **a** Diagram of the integrated stress response (ISR) and the actions of small-molecular inhibitors of ISR. **b** Schematic timeline of stress stimuli and inhibitor treatments, and induction of luciferase reporters. **c** G3BP immunofluorescence of reporter cells treated with arsenite and MG132, without or with pretreatment of ISRIB and PERKi. Stress granules were detected with G3BP antibody. **d** Quantification of stress granule numbers per cell in **c**. More than 100 cells were counted in each condition. ***$P < 0.0005$, two-tailed *t*-test. **e** Fold change of relative RAN translation vs. AUG translation in bicistronic reporters under arsenite (left) and MG132 (right) stimuli, without or with pretreatment of ISRIB or PERKi compounds targeting the eIF2α pathway. NLuc signals were normalized to FLuc. **$P < 0.005$, ***$P < 0.0005$, two-tailed *t*-test. **f** Schematic timeline of cDNA transfection and inhibitor treatment (as in **h**), and induction of luciferase reporters. **g** Relative expression of C9R-NLuc vs. AUG-FLuc reporters upon transfection of eIF2α wild-type or S51D mutant were compared with negative control. NLuc signals were normalized to FLuc in each sample. *$P < 0.05$, **$P < 0.005$, two-tailed *t*-test. **h** Fold change of bicistronic RAN translation reporter expression upon eIF2α-S51D transfection, without or with ISRIB treatment. *$P < 0.05$, **$P < 0.005$, two-tailed *t*-test. Data are mean ± SEM from three biological replicates

translation efficiency (determined by normalizing the protein levels (Luciferase activities) to the corresponding RNA levels) from the bicistronic splicing reporter was ~15–35% of the cap-dependent translation from the monicistronic reporter (Fig. 5d), consistent with the relative translation level of the transfected RNAs with or without functional 5′-cap synthesized in vitro (Fig. 1l). The bicistronic reporter showed lowest translation level (~5% of monocistronic reporter), probably because the position of the repeats on a transcript influences its activity.

Both the spliced intron RNA and the unspliced pre-mRNA contain the GGGGCC repeats and could be templates for the RAN translation of C9R-NLuc. Recognizing that both pre-mRNAs and excised introns are generally retained in the nucleus, while translation is widely accepted to be in the cytoplasm, we therefore fractionated cells to separate nucleus and cytoplasm and examined the locations of repeat-containing RNAs. We designed primers across the exon–intron junctions to measure pre-mRNAs, and primers for NLuc to amplify from both pre-mRNA and excised intron (Fig. 5a). GAPDH pre-mRNA was highly enriched in the nucleus, and the mitochondria 12S RNA MTRNR1 was predominantly in cytoplasm, confirming the

successful fractionation (Fig. 5e). As expected, the C9R-NLuc reporter pre-mRNA was predominantly localized in the nucleus, similar to GAPDH pre-mRNA (Fig. 5e, right). For the spliced reporter mRNA, the ratio in cytosol vs. nuclear fraction was close to 1 (Fig. 5e). Surprisingly, NLuc amplicon-containing RNAs accumulated in the cytoplasm to much higher levels than did the pre-mRNAs (Fig. 5e, right). This evidence suggests that the spliced repeat-containing intronic RNA, but not the unspliced pre-mRNA, is highly likely to serve as the main template of RAN translation after its export to the cytoplasm.

We further applied the translating ribosome affinity purification (TRAP) method[60, 61] to test which repeat-containing RNAs are associated with polyribosomes. Retroviral transduction was used to stably express GFP-tagged RPL10A in our reporter cells. We then used GFP immunoprecipitation to purify the ribosome complexes together with the bound RNA substrates from the cytosolic fraction (Fig. 5f). The intronic NLuc RNA was associated with ribosomes, while the unspliced pre-mRNA was depleted after immunoprecipitation of ribosome complexes (Fig. 5g). This result strongly indicates that the spliced repeat-

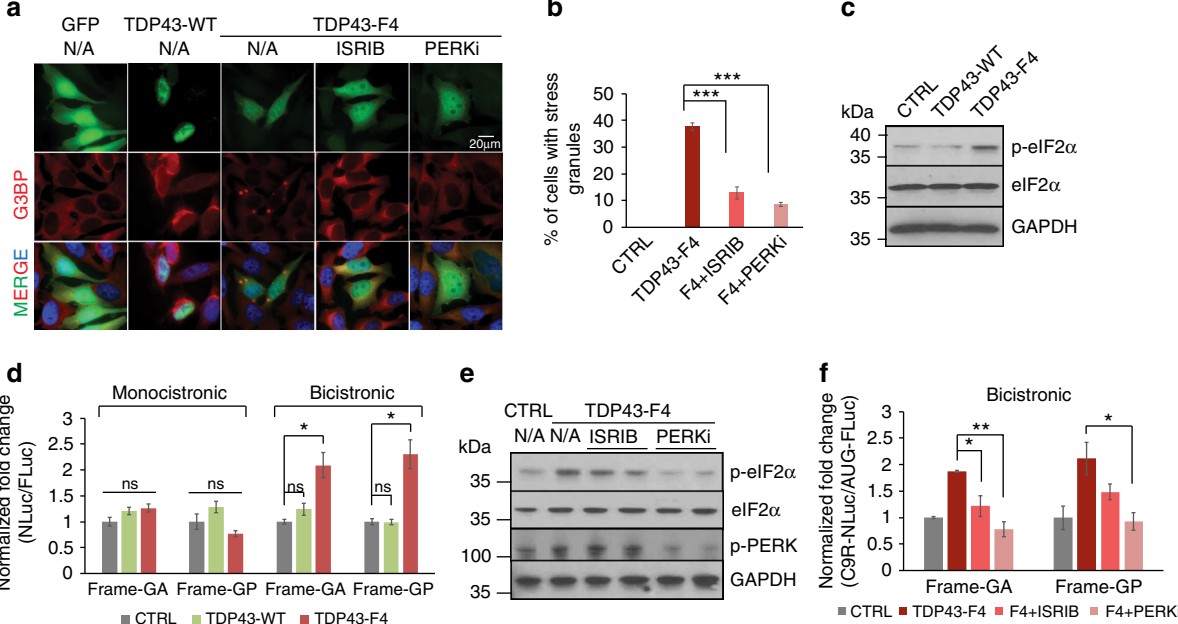

**Fig. 4** TDP-43 prion-like domain promotes RAN translation through phospho-eIF2α. **a** GFP fluorescence and G3BP immunofluorescence of HeLa Flp-In cells transfected with GFP control, TDP43-WT, or TDP43-F4, without or with the treatment of ISRIB or PERKi. **b** Quantification of stress granule numbers per GFP-positive cell in **a**. More than 100 cells were counted in each condition. \*\*\**P* < 0.0001, two-tailed *t*-test. **c** Immunoblotting using antibody recognizing phospho-eIF2α showed its upregulation with TDP43-F4 expression, compared to GFP and TDP43-WT controls. GAPDH and total eIF2α were blotted as internal control. **d** After 1 day of transfection cells were induced to express translation reporters by doxycycline, and luciferase activities were measured after another 24 h. NLuc signals were normalized to FLuc in each sample and the relative expression was compared to GFP transfection control. \**P* < 0.05, two-tailed *t*-test. **e** Cells were treated with ISRIB or PERKi after transfection with TDP43-F4. Immunoblotting with antibodies recognizing phospho-eIF2α, eIF2α, phospho-PERK, and GAPDH. **f** Fold change of bicistronic RAN translation reporter expression upon TDP43-F4 expression, without or with the treatment of ISRIB or PERKi. \**P* < 0.05, \*\**P* < 0.005, two-tailed *t*-test. Data are mean ± SEM from three biological replicates

containing intronic RNA, but not the unspliced pre-mRNA, is the main substrate of RAN translation.

**Translation of intronic repeats is enhanced by ISR.** We next tested whether the translation of the intronic repeats was dependent on the 5′-cap. Knockdown of eIF4E by siRNA reduced the AUG-FLuc to about half, but had little effect on the C9R-NLuc expression (Fig. 6a, b). Inhibition of the mTOR signaling pathway by Torin 1 also only reduced the AUG-FLuc translation and did not change C9R-NLuc translation (Figs. 6c, d), as expected for cap-independent translation. The intronic C9R-NLuc translation was upregulated by stress inducers, arsenite and MG132 (Fig. 6e), without changes in splicing and RNA expression levels of the NLuc reporter (Supplementary Fig. 3b,c). Consistent with our biscistronic reporters (Fig. 3e), the eIF2α signaling pathway inhibitors ISRIB and PERKi reduced stress-induced translation elevation of the intronic repeats (Fig. 6e). Expression of eIF2α S51D mutant also enhanced C9R-NLuc expression and this effect was inhibited by ISRIB (Fig. 6f). Collectively, these data indicate that when the (GGGGCC)70 repeats are localized in the *C9ORF72* intron, the resultant RAN translation is predominantly from the excised intron, which is cap-independent and is upregulated by stress through the phospho-eIF2α signaling pathway.

## Discussion

Using a series of inducible reporter cell lines, our work has provided insights on the mechanism of *C9ORF72* (GGGGCC)n RAN translation in vivo, both at basal level and in response to stress. Our data indicate that RAN translation of *C9ORF72* sense repeats can initiate with and without a 5′-cap, with the location and context of the repeat expansion influencing the initiation

mechanism and efficiency. In the context of RNAs derived for the *C9ORF72* gene, we demonstrate that RAN translation of GGGGCC repeats arises mostly from the uncapped, spliced intronic RNA. Hence, translation initiation of the *C9ORF72* repeats not only bypasses the need for an AUG start codon, but also does not require the 5′-cap or the cap-binding protein eIF4E. This cap-independent translation is similar to the IRES-mediated pathway used widely by many virus RNAs and a handful of cellular RNAs[42, 43]. Previous studies have reported that IRES translation can be regulated by IRES-transacting factors that do not have known function in canonical translation, especially under different physiological and pathological conditions[42, 43]. RBPs have been shown to modulate IRES translation through altering the affinity between RNA structures and translation factors[42]. Recognizing those precedents, our finding of cap-independent translation of C9ORF72 repeats strongly suggests the possibility that specific (as yet unidentified) factors are recruited by the GGGGCC repeats to facilitate ribosome assembly and translation initiation without affecting canonical translation.

Our evidence revealed that the cap-independent RAN translation of the repeats in the excised *C9ORF72* first intron is very likely to be the predominant mechanism in C9ALS/FTD patients, as the initiation rate is quite efficient, ~15–35% of a 5′-capped RNA with the same repeat length. Furthermore, recognizing that previous evidence has reported splicing of the *C9ORF72* first intron is either unaffected[62] or only modestly affected[63] by the repeat expansion, the most abundant repeat-containing RNA is probably the excised intron, which is usually not co-transcriptionally capped at the 5′ end. Our data showed that the unspliced pre-mRNA is not exported to, or accumulated within the cytoplasm and is not associated with ribosomes. Therefore, cap-independent translation of repeat-containing

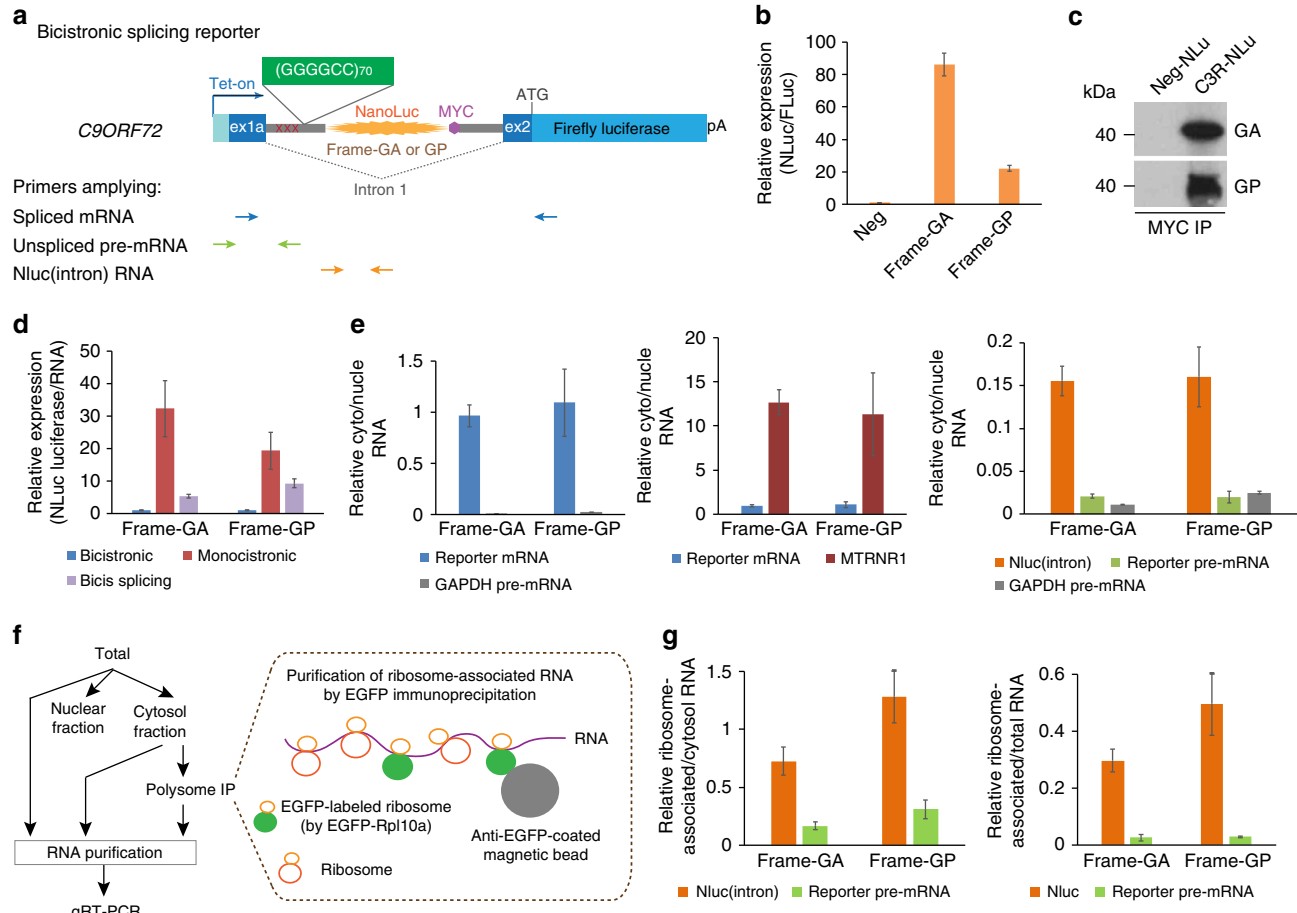

**Fig. 5** RAN translation of GGGGCC repeats from *C9ORF72* spliced intron. **a** Schematic of bicistronic splicing dual-luciferase reporters with *C9ORF72* exons and intron. C9R-NLuc reporter is located in intron 1 and AUG-FLuc reporter is in exon 2 fused with the original C9ORF72 start codon. The positions of primers amplifying different RNA products were labeled. **b** Relative RAN translation products from Frame-GA and Frame-GP were compared to no-repeat control, after 24 h induction in HeLa Flp-In cells. NLuc signals were normalized to FLuc in each sample. **c** Immunoprecipitation using MYC antibody from cells expressing Neg-NLuc, C9R-NLuc of frame-GA, or GP in bicistronic splicing reporters, followed by immunoblotting with GA or GP antibody. **d** The relative translation level of C9R-NLuc in monocistronic, bicistronic, and bicistronic splicing reporters. The NLuc luciferase level in each cell line was normalized to the NLuc RNA level, which was quantified by qRT-PCR. **e** After 24 h induction of reporter genes, cells were fractionated to separate nucleus and cytoplasm. The levels of reporter spliced mRNA and unspliced pre-mRNA, Nluc (amplifying excised intron and unspliced pre-mRNA), GAPDH pre-mRNA, and the mitochondria RNA MTRNR1 were measured by qRT-PCR and normalized to GAPDH mRNA in each fraction. The ratio of cytosol/nuclear RNA showed the subcellular distribution of each RNA. The nuclear marker GAPDH pre-mRNA is depleted in cytosol fraction (left) and the cytosol marker MTRNR1 is highly enriched in cytosol fraction (middle). NLuc RNA shows more cytosolic distribution than pre-mRNAs (right). **f** The diagram of experiment design and TRAP methodology to isolate ribosome-associated RNAs. **g** Total, cytosol, and ribosome-associated RNAs were isolated for qRT-PCR. NLuc and reporter pre-mRNA were measured and normalized to GAPDH mRNA in each fraction. The ratios of ribosome-associated/cytosol RNA (left) and ribosome-associated/total RNA (right) show that the Nluc intron RNA is associated with ribosomes, but the pre-mRNA is not. Data are mean ± SEM from three biological replicates

spliced intron is likely to be the major source for DPR production, although we note that this does not exclude the possibility under some circumstances of yet uncharacterized repeat-containing transcripts that contain a 5′ cap and the cap-dependent RAN translation might also contribute to the DPR production. In addition, the GA DPR produced by our C9R-NLuc reporters was consistently accumulated to a higher level than the GP DPR in all the reporter systems. This may be because the translation initiation in frame-GA is more efficient than frame-GP, or poly-GA is more stable than poly-GP, or the translation elongation rate is different, influenced by the relative amount of aminoacyl-tRNAs recognizing different codons.

After splicing, intron-derived RNAs are normally excised into a lariat structure, debranched and degraded by exonucleases

intranuclearly[41]. Correspondingly, they cannot have the 5′-m7G cap and poly(A) tail added co-transcriptionally, and are generally thought not to be exported to the cytoplasm. However, our data have identified export of the repeat-containing, spliced *C9ORF72* first intron from the nucleus to the cytoplasm. There are multiple prior examples of excised introns of cellular RNAs exported to the cytoplasm[64, 65], as well as several viruses that are known to promote export of intron-containing viral RNA transcripts through selective interaction between structured cis-acting RNA elements and cellular nuclear export factors[66]. A recent work identified that SRSF1 binds GGGGCC repeats and mediates NXF1-dependent nuclear export[67], indicating the importance of repeat-binding RBPs in determining the distinct molecular destination of RNA repeats.

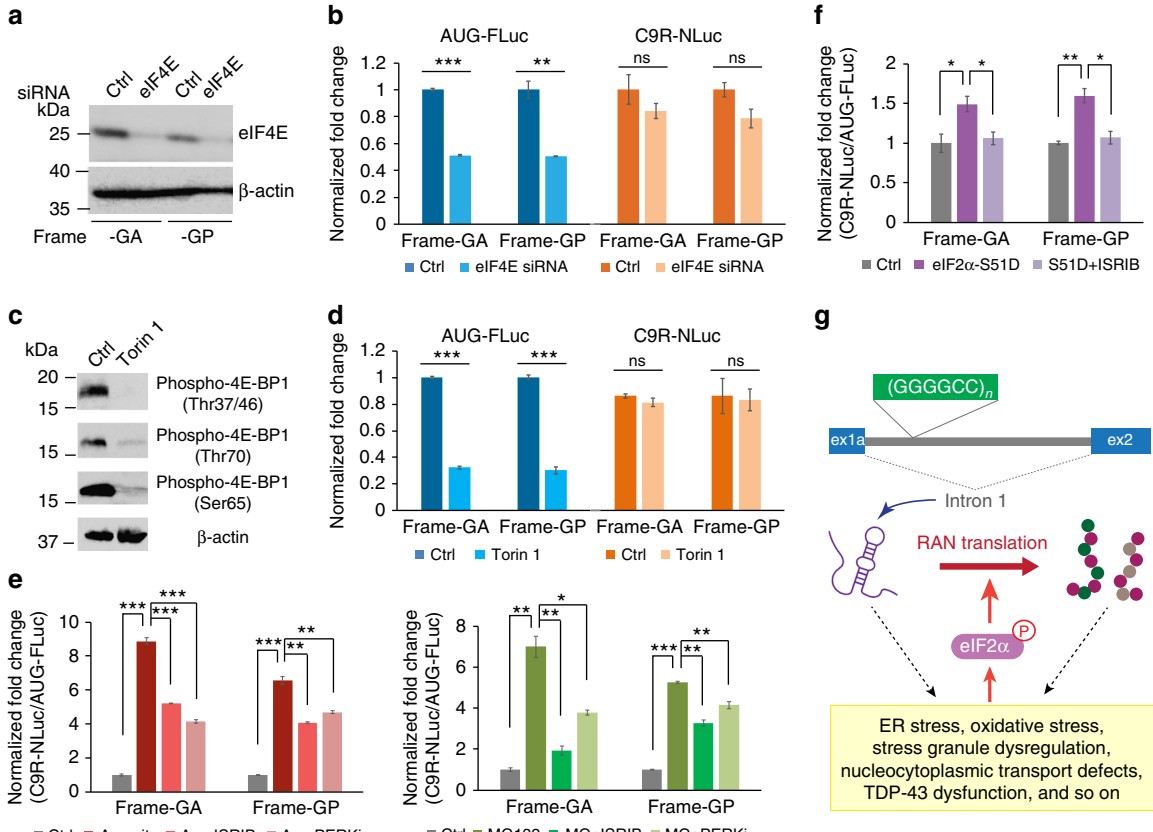

**Fig. 6** Translation of intronic repeats does not require 5'-cap and can be upregulated by stress. **a** HeLa reporter cells were transfected with non-targeting siRNA or siRNA against cap-binding protein eIF4E. Immunoblotting of eIF4E showed the knockdown efficiency. β-actin was blotted as internal control. **b** Expression of AUG-FLuc translation (left) and C9R-NLuc RAN translation (right) reporters in presence of non-targeting siRNA or siRNA against eIF4E. **P < 0.005, ***P < 0.0005, two-tailed t-test. **c** Bicistronic splicing reporter cells were treated with mTOR inhibitor Torin 1 for 24 h. Immunoblotting of phospho-4E-BP1 using antibodies recognizing different phosphorylation sites. β-actin was blotted as internal control. **d** Expression of AUG-FLuc and C9R-NLuc in bicistronic splicing reporter cells with the treatment of mTOR inhibitor Torin 1. **P < 0.005, ***P < 0.0005, two-tailed t-test. **e** Fold change of relative RAN translation vs. AUG translation of the bicistronic splicing reporters under arsenite (left) and MG132 (right) stimuli, without or with pretreatment of ISRIB or PERKi. NLuc signals were normalized to FLuc in each sample. *P < 0.05, **P < 0.005, ***P < 0.0001, two-tailed t-test. **f** Fold change of RAN translation upon eIF2α-S51D expression, without or with the treatment of ISRIB. *P < 0.05, **P < 0.005, two-tailed t-test. **g** Diagram for a feedforward loop with C9ORF72 (GGGGCC)$_n$ RAN translation being enhanced by initial repeat-mediated toxicity through eIF2α phosphorylation pathway. Data are mean ± SEM from three biological replicates

Stress responses and stress granule dysfunction have been increasingly associated with disease pathogenesis of ALS/FTD[45]. During ISR, there is global translation reduction accompanied by increased translation of selective RNAs that are believed to be important for cell survival under stress[54]. eIF2α phosphorylation, whose selective elevation within neurons has been reported in multiple neurodegenerative diseases[61, 68, 69], has also been reported to be a critical hub for the control of neuronal synaptic plasticity and memory consolidation[70, 71]. We have demonstrated here that activation of this signaling pathway stimulates RAN translation of C9ORF72 repeats. It is not clear how eIF2α phosphorylation enhances RAN translation. One possibility is that when global translation is reduced, some of the rate-limiting steps or components become more available to the unconventional initiation, therefore increasing the RAN translation rate. However, inhibition of the mTOR pathway decreases canonical translation but cannot enhance RAN translation, suggesting that the availability of translation machinery is not the only reason for upregulation of cap-independent translation. In the adult-onset progressive neurodegenerative diseases, aging factors and internal stress stimuli originally arising from the C9ORF72 repeat-mediated toxicity (including DPR-induced oxidative stress[24],

stress granule dysfunction[26, 28], altered ER homeostasis[72, 73], and TDP-43 mis-localization[11]) may trigger a feedforward loop to upregulate RAN translation and generate an increasing amount of DPRs that exert more toxicity and eventually lead to neuronal dysfunction, degeneration, and ultimately death, thereby driving relentless disease progression (Fig. 6g). A method perturbing this loop might reduce or delay neurodegeneration and hold therapeutic promise in C9ORF72-ALS/FTD.

## Methods

**Plasmids**. For monocistronic reporters, the FLuc-coding sequence was cloned into pBABE (puro) via BamHI and SalI. To generate NLuc reporter, multiple stop codons in all three reading frames were introduced after the CMV promoter of the pcDNA5-FRT-TO vector by QuickChange mutagenesis. The NLuc-coding sequence lacking ATG start codon with C-terminal MYC tag and preceding C9ORF72 intron sequence (before the repeats) was PCR-amplified and cloned into the modified pcDNA5-FRT-TO vector by HindIII and XhoI. About (GGGGCC)$_{70}$ repeat sequence was cloned between intron sequence and NLuc by NotI and blunt end ligation. The fusion with different reading frames was achieved by inserting one nucleotide shift during PCR amplification of the NLuc gene. For bicistronic reporters, the FLuc-coding sequence containing multiple stop condons in three reading frames was cloned into pcDNA5-FRT-TO vector via KpnI and EcoRV. The C9R-NLuc and the Neg-NLuc were cut out from the monocistronic reporter by HindIII and XhoI, DNA blunting the HindIII end, and cloned after FLuc via EcoRI

and XhoI sites. For bicistronic splicing reporter, the sequences including C9ORF72 exon 1a, exon 2, and around 200nt intron 1 from each exon–intron junction were synthesized (Genewiz) and cloned into pcDNA5-FRT-TO vector via NheI and XhoI sites. We included HindIII and BamHI sites inside the synthesized sequence close to the repeat expansion location and cloned in the C9R-NLuc and Neg-NLuc from the monocistronic reporter. For in vitro transcription, the C9R-NLuc and the Neg-NLuc were cut out from the monocistronic reporter by HindIII and XhoI and subcloned in pcDNA3.1+ downstream of the T7 promoter. The wild-type eIF2α cDNA was subcloned into pcDNA 3.1 vector with N-terminal Flag tag via BamHI and XhoI sites. The S51D mutation was introduced by QuickChange mutagenesis. The TDP-43 F4 construct was acquired from Addgene (#28197). The wild-type TDP-43 construct was generated by replacing the F4 fragment with full-length cDNA via XhoI and BamHI sites. The ATF4 IRES vector was engineered by inserting the sequence of ATF4 5′UTR with the first 32 codons of ATF4[56] into the bicistronic reporter between FLuc and NLuc, fused with the NLuc gene at the EcoRV site by In-Fusion cloning. To generate the retroviral vector expressing Rpl10a, the Rpl10a cDNA was cloned into pEGFP-C1 via XhoI and BamHI, and the EGFP-Rpl10a was subsequently cloned into the pBABE vector via EcoRI and BamHI.

**Cell culture and transfection**. HeLa Flp-In cells were grown in DMEM supplemented with 10% (v/v) FBS, 100 U ml$^{-1}$ penicillin and 100 μg ml$^{-1}$ streptomycin. HeLa stable cell lines expressing various translation reporters were generated as described before[46]. For transfection experiment, the reporter gene expression was induced with 2 μg ml$^{-1}$ doxycycline for 24 h prior to sample collection. For mTOR inhibition, cells were treated with Torin 1 at 1 μM for 24 h. For stress stimuli experiment, cells were treated with sodium arsenite at 200 μM, or MG132 at 10 μM, together with reporter induction for 6 h. ISRIB (Sigma) and GSK260641 (PERKi, Sigma) were added to cells 24 h before harvest at 0.5 and 1 μM. TransIT-LT1 (Mirus) was used to transfect plasmids; Lipofectamine RNAiMAX (Invitrogen) was used to transfect siRNAs. siGENOME eIF4E siRNA and siGENOME Non-Targeting siRNA (GE Dharmacon) were transfected at 25 nM. For transfection experiments, plasmids were transfected 48 h and siRNAs were transfected 72 h prior to sample collection. The FLuc siRNA (5′-GGACGAGGACGAGCACUUC-3′) was transfected at 50 nM 24 h after reporter gene induction. 293 *Phoenix* cells were used for retrovirus packaging. Virus-infected cells were selected using puromycin (0.5 μg ml$^{-1}$). EGFP-Rpl10a-expressing cells were further confirmed by fluorescence-activated cell sorting. Three wells of biological replicates were prepared at each condition for all dual-luciferase reporter experiments. The NLuc and FLuc luciferase activities were measured by Nano-Glo Dual Luciferase Assay (Promega) on Tecan Infinite 200 PRO. NLuc levels were normalized to FLuc, or both were normalized to total protein amounts for each sample. Protein lysates were quantified by BCA Assay (ThermoFisher Scientific).

**In vitro transcription**. The pcDNA3.1-NLuc constructs were linearized with XhoI digestion, and served as the DNA template to synthesize RNA using MEGAscript T7 transcription kit (Ambion), in presence of either normal m$^7$GpppG cap analog or non-functional ApppG cap analog (New England Biolabs). Synthesized RNAs were transfected into cells by Xfect RNA transfection kit (Clontech), and samples were collected after 24 h.

**Immunoprecipitation**. Dynabeads Protein G was washed twice with PBST and incubated with MYC antibody (Sigma, 05–724) for 1 h at room temperature. The beads were washed twice in PBST, and once in immunoprecipitation (IP) lysis buffer (0.3% (v/v) NP-40, 200 mM NaCl, 50 mM Tris, pH 7.4, 1 mM 1,4-Dithiothreitol (DTT)), 0.1 mM EDTA, and protease-inhibitor cocktail). HeLa cells induced to express various reporter genes were lysed in IP lysis buffer, and the DNA sheared by sequential passage through a syringe with 22 and 26 G needles, three times each. The lysates were clarified by centrifugation at 13,000*g* for 20 min at 4 °C. The antibody-coated beads were incubated with the cleared lysates and incubated for 4 h at 4 °C. After washing three times in lysis buffer, the beads were resuspended in SDS-containing gel sample buffer and electrophoresed on an SDS-PAGE gel.

**Translating ribosome affinity purification**. Cells expressing EGFP-Rpl10a were collected and lysed in gentle lysis buffer (20 mM HEPES KOH, pH7.4, 10 mM KCl, 3 mM MgCl$_2$, 0.3%(v/v) NP-40, 0.1 mM EDTA, 1 mM DTT, 100 μg ml$^{-1}$ cycloheximide, protease-inhibitor cocktail, and 200 U ml$^{-1}$ RNase inhibitor). Lysates were centrifuged at 2300*g* for 5 min at 4 °C to separate cytosol and nuclear fractions. The supernatant was transferred to a new tube and the KCl concentration was adjusted to 150 mM. The lysates were centrifuged again at 13,000*g* for 20 min at 4 °C. The GFP antibody-coated Protein G Dynabeads were incubated with the cleared lysates overnight at 4 °C with rotation. Beads were subsequently washed five times with high-salt polysome wash buffer (20 mM HEPES pH 7.4, 350 mM KCl, 5 mM MgCl$_2$, 1 mM DTT, 1% Nonidet P-40, and 100 μg mL$^{-1}$ cycloheximide). Trizol was directly added to the beads to extract ribosome-bound RNAs. Total and cytosol RNAs were also extracted, respectively, from the aliquot lysates before and after cell fractionation.

**Immunofluorescence and immunoblotting**. Cells were fixed with 4% (v/v) paraformaldehyde in phosphate-buffered saline (PBS) for 20 min, permeabilized in 0.2% (v/v) Triton X-100 for 5 min, blocked in 1% bovine serum albumin, and 2% goat serum for 30 min, incubated with primary antibodies for 1 h, washed with PBS, and finally incubated with Alexa Fluor 546-conjugated secondary antibodies (ThermoFisher Scientific). Nuclei were counterstained with 4′,6-diamidino-2-phenylindole (DAPI). Cells were imaged with a fluorescence microscope (Zeiss Axiophot). For immunoblotting, goat anti-mouse or anti-rabbit IgG horseradish peroxidase-conjugated antibody (GE Healthcare) was used along with chemiluminescent detection reagents (Thermo Scientific). The primary antibodies included eIF4E (Bethyl, A301-154A, 1:1000), phospho-eIF2α (Cell Signaling, 9721, 1:1000), eIF2α (Cell Signaling, 9722, 1:1000), phospho-PERK (Santa Cruz, sc-32577, 1:1000), G3BP (BD, 611126, 1:300 for IF), GAPDH (Cell Signaling, 97166, 1:1000), poly-GA (Rb4334, 1:1000), poly-GP (Rb4336, 1:1000), FLAG (Sigma, F1804, 1:500), phospho-4E-BP1 Thr37/46 (Cell Signaling, 2855, 1:1000), phospho-4E-BP1 Thr70 (Cell Signaling, 9455, 1:1000), phospho-4E-BP1 Ser65 (Cell Signaling, 9451, 1:1000), β-actin (Cell Signaling, 3700, 1:1000), and GFP (Memorial Sloan Kettering Cancer Center Monoclonal Antibody Core Facility, Htz-GFP19C8). Full scans of immunoblotting images are provided in Supplementary Fig. 4.

**RNA isolation, qRT-PCR and RT-PCR**. To isolate total RNA from cells, Trizol (Invitrogen) and treatment with RQ1 DNase I (Promega) was used. For first-strand cDNA synthesis, random hexamers were used with High-capacity cDNA reverse transcription kit (Applied Biosystems).

All qRT-PCR reactions were performed with three biological replicates for each group and two technical replicates using the iQ SYBR green supermix (Bio-Rad) on the CFX96 real-time PCR detection system (Bio-Rad). The data were analyzed using the CFX96 optical system software (Bio-Rad; version 1.1). Expression values were normalized to GAPDH mRNA. Intergroup differences were assessed by two-tailed Student's *t*-test. Primer sequences are listed in Supplementary Table 1.

For cell fractionation, cells were lysed in gentle lysis buffer (20 mM Tris pH 7.4, 10 mM NaCl, 3 mM MgCl$_2$, 0.3% (v/v) NP-40). Nuclei were pelleted at 2300*g* for 5 min at 4 °C, and supernatant (cytosol fraction) was transferred to a new tube. The nuclei were re-suspended in gentle lysis buffer and spun down again to collect the pellet (nuclear fraction). Trizol was directly added to the two fractions for subsequent RNA extraction.

**Data availability**. The data that support the findings of this study are available from the corresponding author upon reasonable request.

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

## Acknowledgements

We thank Dr. Leonard Petrucelli and Dr. Tania Gendron for suggestions on dipeptide measurement. We thank Sun lab members for helpful discussion. This work was supported by grants to S.S. from the NIH (R00NS091538), Target ALS, and the Robert Packard Center. J.J. is a recipient of career development grant from Muscular Dystrophy Association (479769) and was supported by postdoctoral training grant (T32 AG00216) and postdoctoral fellowship (F32 NS087842) from the NIH.

## Authors contributions

W.C. and S.S. designed the research and wrote the manuscript. W.C., S.W., A.A.M., C.F., A.M., and F.X. performed experiments. L.R.H., R.L.-G., K.D., J.J., and D.W.C. provided key reagents.

## Additional information

**Competing interests:** The authors declare no competing financial interests.

