## [Peer Review File · Nature Communications]

Reviewers' comments:

Reviewer #1 (Remarks to the Author):

C9ORF72 GGGGCC repeat expansion is the frequent cause of amyotrophic lateral sclerosis (ALS) and frontotemporal dementia (FTD). In this manuscript, Cheng et al. used inducible reporter genes to examine the mechanism of repeat-associated non-AUG (RAN) translation of (GGGGCC)_n-containing RANs in vivo under normal and stress conditions. Their results indicate that this translation can be initiated by cap-independent (IRES-like) mechanism when the repeats are derived from RNAs located in the first C9ORF72 intron. Furthermore, phosphorylation of eIF2 α , which is the major feature of an integrated stress response, was found to stimulate cap-independent RAN translation. Induction of eIF2 α phosphorylation and stress granule formation by TDP-43 prion-like domain upregulated RAN translation. This stimulation was reduced by compounds inhibiting eIF2 α phosphorylation. Thus, these compounds might be useful for the therapy of C9ORF72-ALS/FTD.

This study is important for our understanding of the mechanism of RAN translation of C9ORF72 GGGGCC repeat expansion and its regulation under stress conditions, and the etiology of ALS and FTD. However, it is very surprising that the (GGGGCC)_n-containing intron is entering the cytoplasm, and therefore they need provide more solid proof to support their conclusion. Also, the conclusion about cap-independence of RAN translation must have more support, as detailed below.

Points:

1. Fig. 1. As for many cellular IRESs, data from the bicistronic DNA expression is prone to misinterpretation, as a fraction of monocistronic mRNA could be transcribed from cryptic promoters (ref.1). A good control would be to co-express a siRNA that targets the first (F-Luc) cistron coding sequence. If the siRNA causes no reduction in the second cistron expression, the most likely scenario is that this expression is driven by unintended monocistronic mRNAs (ref.2).
2. Fig. 2b and Fig. 3. To confirm analogy between the GGGGCC repeat and IRES with respect to their stimulation by eIF2 α phosphorylation they need to provide a control using a well-established viral IRES (e.g., polio/EMCV or HCV IRES).

3. Fig. 5. The use of “gentle” cell lysis buffer may not prevent some leakage of RNA from the nucleus to the cytoplasm. Could they also show that the GGGGCC repeat-containing intronic RNA is present in the cytoplasm by FISH analysis?
4. What is the possible mechanism of stimulation of RAN translation by eIF2 α phosphorylation? Could this be just a consequence of global translation inhibition and reduced competition from the bulk of cellular mRNAs for the translation machinery? This should be discussed.

References

1. Jackson, R.J. (2013) The current status of vertebrate cellular mRNA IRESs. *Cold Spring Harb Perspect Biol* 2013;5:a011569.
2. Van Eden, M.E., Byrd, M.P., Sherrill, K.W. and Lloyd, R.E. (2004) Demonstrating internal ribosome entry sites in eukaryotic mRNAs using stringent RNA test procedures. *RNA*, 10, 720-730.

Reviewer #2 (Remarks to the Author):

This manuscript addresses problems associated with microsatellite repeats, in particular the GGGGCC repeat. There are two primary findings. The first is that expression from mRNAs containing these repeats is increased under conditions of stress (arsenite, MG 132, TDP-43, etc.). The second finding is that the majority of the mRNA that enters the cytosol appears to come from the intron and thus is not capped (and one assumes, not polyadenylated) although chemical confirmation of the ends is not presented.

Specific concerns

1. A major concern is the lack of absolute numbers for the expression of capped and non-capped mRNAs as these are buried by showing “relative ratio’s”. It is highly likely that the level of expression from an uncapped mRNA is going to be 1% or less than that of a capped mRNA which will compete better for the translational machinery. For example, what is the absolute value for the level of expression of AUG-Fluc in Figure 1, panel I to the level of expression of GA or GP in both the monocistronic and bicistronic constructs in the same panel?
2. A concern when comparing different cell treatments is the level of the mRNAs that are used to direct expression. Thus, in Figure 1 there is no representation that indicates that the level of mRNA transcription (and subsequent processing) has yielded equivalent amounts of mRNA for translation.
3. Given the whole cell nature of the experiment, it is surprising that when the siRNA against eIF4E is used, there is only a 50% reduction in expression from AUG-Fluc when it appears the

reduction in eIF4E is at least 80% if not more.

4. It is not clear why apparently contradictory results are obtained in Figure 2 when comparing the mono- vs. the dicistronic constructs and their response to stress. This reviewer would anticipate that the absolute level of expression is greater in the monocistronic reporter and thus, the induction that is seen in the dicistronic reporter may represent a much reduced level of expression. However, the monocistronic version is more representative of the actual mRNA.

5. Although the authors monitor the phosphorylation of eIF2, it would also be of value to know how much (%) of the eIF2 is phosphorylated. Second, there is also the possibility that the stressing agent may influence the mTOR pathway and so the authors should also monitor the levels of phosphorylation of 4E-BP.

6. The data in Figure 5 are consistent with the intron as serving as a major source of the mRNA. However, if even only a small portion of the repeat appeared in a capped transcript in the cytosol, if its translational efficiency was much greater, then it still might be the primary cause for the repeat disease (i.e. if there is 10 times more intron mRNA but this mRNA is translated at only 1% of the efficiency of the capped transcript, then there would be 10-fold more expression from the capped transcript).

7. In Figure 7, panel C the level of expression is increased by about 30% with the introduction of the phospho-mimic, eIF2-S51D. But the level of induction with arsenite in panel B is about 8-fold. Why is there such a difference (or what is the level of expression of eIF2-S51D relative to the endogenous eIF2)?

8. In the Discussion and elsewhere, is it possible that the relative level of GA vs. GP synthesis is influenced by the relative amount of tRNA in the cell that recognizes either the GCC codon (Ala) or the CCG codon (Pro)?

Minor concern

1. Discussion – “From a series inducible reporter cells we determined the mechanism of C9orf72 GGGGCC RAN translation in vivo” The authors have provided insights into the mechanism of RAN translation but they have not determined the mechanism.

Cheng et al, submitted to Nat. Comm., *C9ORF72 GGGGCC Repeat-Associated Non-AUG Translation is upregulated upon stress through eIF2 α phosphorylation*

Response to Reviewers' comments:

Reviewer #1 (Remarks to the Author):

C9ORF72 GGGGCC repeat expansion is the frequent cause of amyotrophic lateral sclerosis (ALS) and frontotemporal dementia (FTD). In this manuscript, Cheng et al. used inducible reporter genes to examine the mechanism of repeat-associated non-AUG (RAN) translation of (GGGGCC) n -containing RANs in vivo under normal and stress conditions. Their results indicate that this translation can be initiated by cap-independent (IRES-like) mechanism when the repeats are derived from RNAs located in the first C9ORF72 intron. Furthermore, phosphorylation of eIF2 α , which is the major feature of an integrated stress response, was found to stimulate cap-independent RAN translation. Induction of eIF2 α phosphorylation and stress granule formation by TDP-43 prion-like domain upregulated RAN translation. This stimulation was reduced by compounds inhibiting eIF2 α phosphorylation. Thus, these compounds might be useful for the therapy of C9ORF72-ALS/FTD.

This study is important for our understanding of the mechanism of RAN translation of C9ORF72 GGGGCC repeat expansion and its regulation under stress conditions, and the etiology of ALS and FTD. However, it is very surprising that the (GGGGCC) n -containing intron is entering the cytoplasm, and therefore they need provide more solid proof to support their conclusion. Also, the conclusion about cap-independence of RAN translation must have more support, as detailed below.

Points:

1. Fig. 1. As for many cellular IRESs, data from the bicistronic DNA expression is prone to misinterpretation, as a fraction of monocistronic mRNA could be transcribed from cryptic promoters (ref.1). A good control would be to co-express a siRNA that targets the first (F-Luc) cistron coding sequence. If the siRNA causes no reduction in the second cistron expression, the most likely scenario is that this expression is driven by unintended monocistronic mRNAs (ref.2).

Response: We appreciate the referee raised the possibility of cryptic monocistronic RNA transcription from the bicistronic reporter, and thanks for the suggested experiment to rule out this possibility. We now used siRNA targeting FLuc cistron coding sequence and observed the NLuc cistron expression was also decreased (Supplementary Fig.1a,b). This proves that the NLuc cistron is in the same transcript as FLuc, and its translation is cap-independent.

2. Fig. 2b and Fig. 3. To confirm analogy between the GGGGCC repeat and IRES with respect to their stimulation by eIF2 α phosphorylation they need to provide a control using a well-established viral IRES (e.g., polio/EMCV or HCV IRES).

Response: Thank you for raising this point. We now included data for a well-established cellular IRES-translation activated by stress, the ATF4 5'UTR containing a highly structured region responsible for internal ribosome entry (Chan, et al. 2013). We observed similar changes induced

by arsenite stimuli and eIF2 α S51D expression, which can also be inhibited by ISRIB treatment (Supplementary Fig.2f).

Chan, C.P., Kok, K.H., Tang, H.M., Wong, C.M. & Jin, D.Y. Internal ribosome entry site-mediated translational regulation of ATF4 splice variant in mammalian unfolded protein response. *Biochim Biophys Acta* **1833**, 2165-75 (2013)

3. Fig. 5. The use of “gentle” cell lysis buffer may not prevent some leakage of RNA from the nucleus to the cytoplasm. Could they also show that the GGGGCC repeat-containing intronic RNA is present in the cytoplasm by FISH analysis?

Response: We agree the biochemical fractionation is never 100% absolutely clean. It is more important to compare the relative enrichment of RNAs in the cytosol versus nuclear fractions. The nuclear and cytosol markers showed the fractionation purity in our assay. At this condition, we clearly observed differences between intronic NLuc RNA and unspliced pre-mRNA, with intronic RNA relatively more enriched in the cytosol fraction, suggesting the predominant nuclear export of intronic RNA but not unspliced pre-mRNA. Unfortunately, the FISH method is not sensitive enough and cannot detect diffused RNA repeats besides big nuclear RNA granules. However, we now applied the translating ribosome affinity purification (TRAP) technique to directly identify the ribosome-associated RNA (Fig.5e). Our data showed that the intronic RNA is associated with ribosomes, but the unspliced pre-mRNA is depleted after the immunoprecipitation of ribosome complexes (Fig.5f). We think this is a stronger evidence to prove that the spliced intronic RNA, but not the unspliced pre-mRNA, is exported to cytoplasm and serves as the major template for RAN translation.

4. What is the possible mechanism of stimulation of RAN translation by eIF2 α phosphorylation? Could this be just a consequence of global translation inhibition and reduced competition from the bulk of cellular mRNAs for the translation machinery? This should be discussed.

Response: Good point! We now included this in Discussion.

References

1. Jackson, R.J. (2013) The current status of vertebrate cellular mRNA IRESs. *Cold Spring Harb Perspect Biol* 2013;5:a011569.
2. Van Eden, M.E., Byrd, M.P., Sherrill, K.W. and Lloyd, R.E. (2004) Demonstrating internal ribosome entry sites in eukaryotic mRNAs using stringent RNA test procedures. *RNA*, 10, 720-730.

Reviewer #2 (Remarks to the Author):

This manuscript addresses problems associated with microsatellite repeats, in particular the GGGGCC repeat. There are two primary findings. The first is that expression from mRNAs containing these repeats is increased under conditions of stress (arsenite, MG 132, TDP-43, etc.). The second finding is that the majority of the mRNA that enters the cytosol appears to come from

the intron and thus is not capped (and one assumes, not polyadenylated) although chemical confirmation of the ends is not presented.

Specific concerns

1. A major concern is the lack of absolute numbers for the expression of capped and non-capped mRNAs as these are buried by showing “relative ratio’s”. It is highly likely that the level of expression from an uncapped mRNA is going to be 1% or less than that of a capped mRNA which will compete better for the translational machinery. For example, what is the absolute value for the level of expression of AUG-Fluc in Figure 1, panel I to the level of expression of GA or GP in both the monocistronic and bicistronic constructs in the same panel?

Response: Thank you for raising this point. We now quantified the relative translation levels of C9R-NLuc in different reporters by normalizing the luciferase activity with RNA expression level measured by qRT-PCR. The C9R-NLuc translation in the bicistronic reporter is about 5% of that in the monocistronic reporter (Supplementary Fig.1c). This is expected as IRES-translation is usually less efficient than canonical translation.

2. A concern when comparing different cell treatments is the level of the mRNAs that are used to direct expression. Thus, in Figure 1 there is no representation that indicates that the level of mRNA transcription (and subsequent processing) has yielded equivalent amounts of mRNA for translation.

Response: We have measured the RNA expression levels of FLuc and NLuc in all the reporters by qRT-PCR and showed there were no changes with the different cell treatment in the original Supplementary Fig.2, and now in Supplementary Fig.2a,e,g.

3. Given the whole cell nature of the experiment, it is surprising that when the siRNA against eIF4E is used, there is only a 50% reduction in expression from AUG-Fluc when it appears the reduction in eIF4E is at least 80% if not more.

Response: We think it is possible that normally the eIF4E amount is more than what’s needed and the left 20% of eIF4E could still exhibit 50% of the function. Another possibility is different substrate RNAs could have different sensitivity to the amount of eIF4E.

4. It is not clear why apparently contradictory results are obtained in Figure 2 when comparing the mono- vs. the dicistronic constructs and their response to stress. This reviewer would anticipate that the absolute level of expression is greater in the monocistronic reporter and thus, the induction that is seen in the dicistronic reporter may represent a much reduced level of expression. However, the monocistronic version is more representative of the actual mRNA.

Response: What we are trying to show is the stress induced changes of each reporter cell line comparing to the individual basal level. We observed that only the cap-independent translation in the bicistronic reporter is upregulated upon stress stimuli comparing to its own basal level. It is true that the translation level of the bicistronic reporter is still lower than the monocistronic reporter. But we don’t think the monocistronic version is more representative, as the GGGGCC repeats is located in the intron and not in the mature mRNA. And this is a big difference from the previously

studied RAN translation, such as the CGG repeats in FXTAS, which is located in the 5' UTR of mRNA. Our study using splicing reporters in Fig.5 and previous studies on patient samples showed that the splicing of this repeat-containing intron is not affected. As far as is known, the predominant RNA species containing the GGGGCC repeats is the spliced intron, which is most likely not capped at the 5'-end. Therefore, we think there are more chances that the bicistronic reporter is more representative for the C9 repeat RAN translation.

5. Although the authors monitor the phosphorylation of eIF2, it would also be of value to know how much (%) of the eIF2 is phosphorylated. Second, there is also the possibility that the stressing agent may influence the mTOR pathway and so the authors should also monitor the levels of phosphorylation of 4E-BP.

Response: Thank you for raising this point. We now looked at the phosphorylation status of 4E-BP1 at multiple sites by immunoblotting. Hyperphosphorylation of 4E-BP1 can inhibit its binding with eIF4E and therefore enhances the cap-dependent translation. We used antibodies recognizing the well-studied phospho-Thr37/46, phospho-Thr70 and Ser65 sites, and observed that only the phospho-Thr37/46 is decreased by Arsenite treatment but the other sites have no changes (Supplementary Fig.2b). It has been shown that the combination of phosphorylation events at all these sites is essential to change the 4E-BP association with eIF4E and the Thr37/46 alone has no effect on translation (Gingras, et al. 1999, 2001). In addition, the mTOR signaling pathway is usually involved in cell growth and proliferation rather than stress responses. Therefore, we believe the stress-induced RAN translation changes in our assay are not due to the mTOR signaling pathway. Unfortunately, due to the different affinity of antibodies recognizing total eIF2 α and phospho-eIF2 α , it is hard to quantify the relative abundance and we didn't find a previous literature showing the percentage of eIF2 α phosphorylation. But we showed that modest expression of the phospho-mimetic mutant of eIF2 α is sufficient to enhance RAN translation (Fig. 3g and Supplementary Fig. 2c).

Gingras, A.C. et al. Hierarchical phosphorylation of the translation inhibitor 4E-BP1. *Genes Dev* **15**, 2852-64 (2001).

Gingras, A.C. et al. Regulation of 4E-BP1 phosphorylation: a novel two-step mechanism. *Genes Dev* **13**, 1422-37 (1999).

6. The data in Figure 5 are consistent with the intron as serving as a major source of the mRNA. However, if even only a small portion of the repeat appeared in a capped transcript in the cytosol, if its translational efficiency was much greater, then it still might be the primary cause for the repeat disease (i.e. if there is 10 times more intron mRNA but this mRNA is translated at only 1% of the efficiency of the capped transcript, then there would be 10-fold more expression from the capped transcript).

Response: We are not clear what could be the source of capped mRNA containing repeats when the repeats are located in the intron, as the excised intron is not in the mature mRNA. Maybe one scenario is the unspliced pre-mRNA regarded as an intron-retention isoform, although it is rarely found in patients in previous studies. But to prove which repeat-containing RNA species are translated, we now applied the translating ribosome affinity purification (TRAP) technique to directly identify the ribosome-associated RNA (Fig.5e). Our data showed that the intronic RNA is associated with ribosomes, but the unspliced pre-mRNA (intron retention) is **depleted** after the

immunoprecipitation of ribosome complex (Fig.5f). We think this evidence strongly indicate that the spliced intronic RNA is exported to cytoplasm and serves as the major template for RAN translation. Even if there were small amount of intron retention isoform generated and exported to cytoplasm, it is not associated with ribosomes and cannot contribute significantly to the RAN translation product. Therefore, we think the cap-independent translation is at least an important contribution (if not the only mechanism) for C9 RAN translation and dipeptide production.

7. In Figure 7, panel C the level of expression is increased by about 30% with the introduction of the phospho-mimic, eIF2-S51D. But the level of induction with arsenite in panel B is about 8-fold. Why is there such a difference (or what is the level of expression of eIF2-S51D relative to the endogenous eIF2)?

Response: Good point! The ectopic expression level of eIF2 α -S51D was very low compared to the endogenous eIF2 α , and it didn't change the overall amount of the cellular eIF2 α . We now included the immunoblotting data for this in Supplementary Fig.2c.

8. In the Discussion and elsewhere, is it possible that the relative level of GA vs. GP synthesis is influenced by the relative amount of tRNA in the cell that recognizes either the GCC codon (Ala) or the CCG codon (Pro)?

Response: Good point! We think it is a very interesting hypothesis and included in Discussion.

Minor concern

1. Discussion – “From a series inducible reporter cells we determined the mechanism of C9orf72 GGGGCC RAN translation in vivo” The authors have provided insights into the mechanism of RAN translation but they have not determined the mechanism.

Response: Thank you for the suggestion. We now corrected the writing.

Reviewers' comments:

Reviewer #1 (Remarks to the Author):

The revised manuscript by Cheng et al and their rebuttal letter satisfactorily address many issues with their original submission. Importantly, the possibility of transcription of the monocistronic NLuc RNA from their bicistronic reporter is now unlikely, given their results with siRNA that targets the first codon sequence (Suppl. Fig. 1a,b). Furthermore, they included a positive control (ATF4 5'UTR, which has been shown to direct internal ribosome entry) to demonstrate similarity between the GGGGCC repeat and IRESs with respect to their stimulation by eIF2 α phosphorylation. Finally, they used the translating ribosome affinity purification (TRAP) technique to identify ribosome-associated mRNA. The data indicate that the spliced GGGGCC repeat-containing intronic RNA, in contrast to the unspliced pre-mRNA, is exported to cytoplasm and serves as a template for RAN translation.

Comments:

1. What remains unclear is the efficiency of translation of uncapped GGGGCC repeat containing mRNA. Comparison of NLuc expression from monocistronic and bicistronic reporters indicated that the cap-independent RAN translation is ~5% of that of the cap-dependent translation (page 7 and Suppl. Fig. 1c). However, I do not think that this is an accurate estimate as the position of the GGGGCC "IRES" in the middle of the bicistronic construct could affect its activity. A better approach would be to compare activities of transfected m7Gppp and Appp-capped monocistronic C9R-NLuc mRNAs.
2. Fig. 2b. Results with monocistronic and bicistronic constructs do not agree with each other, as also noticed by reviewer #2. Although I can accept their explanation, maybe it is worthwhile to delete the monocistronic data, as these may confuse the reader?
3. Fig. 6. eIF4E knockdown by siRNA should be confirmed by Western blotting. In addition, stronger inhibition of cap-dependent translation might be achieved with the use of mTOR inhibitors. These experiments could also substantiate the notion that the stress-induced RAN translation changes are not due to the mTOR signaling pathway (see their response to comment 5, reviewer #2).

Reviewer #2 (Remarks to the Author):

This is an improved manuscript that reports on the expression of polypeptides from the

C9orf72 GGGGCC repeat containing mRNA. The main points established are that the mechanism for initiation seems to follow the IRES-mediated initiation pathway and that the relative level of expression is enhanced by eIF2alpha phosphorylation. It is not clear if this upregulation is due to the loss of competition with the cap-dependent pathway which tends to be dominant in uninhibited cells or not. This reviewer anticipates that an acceptable manuscript can be generated without additional experimentation.

Comments

1. page 6 – line 16 – The fact that translation occurs via an IRES element is NOT an indication that the mRNA is uncapped, but rather that the cap does not play a role in the translational efficiency.
2. Page 10 – line 4 – The authors should make a clear distinction between the normal ATF4 mRNA and the splice variant that contains the IRES (ref. 55). Wild type ATF4 expression is controlled via an upstream ORF while the splice variant is expressed via an IRES element, two very different mechanism but which seem to be run together in this paragraph.
3. The affects seen in Figure 6c seems rather small (50% at best) compared to what is seen in panel d (6 to 8-fold induction) and thus may not truly be reflecting the same phenomena.
4. Figure 2 – Why are the results different depending on whether a mono- or dicistronic mRNA is tested (i.e. inhibition with the monocistronic, stimulation with the bicistronic)? Here and elsewhere, it would be good to also indicate what the absolute value is for the reference (or 1 value) in panel b (and other figures also).

Minor comments

1. page 3, line 17 “...toxic dipeptides ..” should be “toxic polydipeptides”.
2. Page 6, line 6 – poly-A tail should be poly(A) tail
3. Page 7 – line 10 – “ ... and the cap binding protein ...” It is not clear as to whether this means eIF4F or the cap-recognition subunit of eIF4F which is eIF4E.
4. Page 8 – line 16 – While some IRES-containing mRNAs may be either not inhibited or upregulated with eIF2alpha phosphorylation, a number of mRNAs (GCN4, ATF4, etc.) are upregulated when ternary complexes are reduced and this regulation is affected by an upstream ORF(s).
5. Page 13 – line 21 “..the spliced intronic RNA, which is not capped at the 5’ end.” Is there a reference for this or is this assumed with the idea that capping only occurs once with the native transcript? Secondly, in the absence of a cap and a poly(A) tail, it would normally be anticipated that the mRNA here would be short lived and perhaps not even exit the nucleus. Is there an explanation for why this intronic sequence appears in the cytoplasm (i.e. perhaps some unique feature of the repeat?)?

Cheng et al, submitted to Nat. Comm., *C9ORF72 GGGGCC Repeat-Associated Non-AUG Translation is upregulated by stress through eIF2 α phosphorylation*

Response to Reviewers' comments:

We would like to thank both reviewers for all valuable suggestions and comments, which helped us significantly improve our manuscript!

Reviewer #1 (Remarks to the Author):

The revised manuscript by Cheng et al and their rebuttal letter satisfactorily address many issues with their original submission. Importantly, the possibility of transcription of the monocistronic NLuc RNA from their bicistronic reporter is now unlikely, given their results with siRNA that targets the first codon sequence (Suppl. Fig.1a,b). Furthermore, they included a positive control (ATF4 5'UTR, which has been shown to direct internal ribosome entry) to demonstrate similarity between the GGGGCC repeat and IRESs with respect to their stimulation by eIF2 α phosphorylation. Finally, they used the translating ribosome affinity purification (TRAP) technique to identify ribosome-associated mRNA. The data indicate that the spliced GGGGCC repeat-containing intronic RNA, in contrast to the unspliced pre-mRNA, is exported to cytoplasm and serves as a template for RAN translation.

Comments:

1. What remains unclear is the efficiency of translation of uncapped GGGGCC repeat containing mRNA. Comparison of NLuc expression from monocistronic and bicistronic reporters indicated that the cap-independent RAN translation is ~5% of that of the cap-dependent translation (page 7 and Suppl. Fig.1c). However, I do not think that this is an accurate estimate as the position of the GGGGCC "IRES" in the middle of the bicistronic construct could affect its activity. A better approach would be to compare activities of transfected m⁷Gppp and Appp-capped monocistronic C9R-NLuc mRNAs.

Response: Thank you very much for raising this very good point. We now compared translation levels by transfecting cells with *in vitro* synthesized C9R-NLuc and Neg-NLuc RNAs, which contains either m⁷G cap or the non-functional ApppG cap analogue. The cap-independent repeat RNA translation initiation efficiency is about 20-30% of the cap-dependent translation (Fig. 1I), much higher than the bicistronic reporter (~5%), as you pointed out. In particular, this ratio is similar to the relative translation level of the bicistronic splicing reporter, where the repeats are at the 5' end of the spliced and uncapped intron (Fig. 5d). We agree that the position of the repeats in the middle of the bicistronic construct could affect its activity.

2. Fig. 2b. Results with monocistronic and bicistronic constructs do not agree with each other, as also noticed by reviewer #2. Although I can accept their explanation, maybe it is worthwhile to delete the monocistronic data, as these may confuse the reader?

Response: We appreciate your concerns. We now put the data from monocistronic reporter in separate graphs in the supplementary figure, to avoid any confusion.

3. Fig. 6. eIF4E knockdown by siRNA should be confirmed by Western blotting. In addition, stronger inhibition of cap-dependent translation might be achieved with the use of mTOR inhibitors. These experiments could also substantiate the notion that the stress-induced RAN translation changes are not due to the mTOR signaling pathway (see their response to comment 5, reviewer #2).

Response: Thank you for raising this point. We now included western blotting of eIF4E knockdown in Fig.6a. And thanks for the suggestion to inhibit cap-dependent translation using mTOR inhibitors. We indeed achieved stronger inhibition on cap-dependent translation and observed no change on C9R-NLuc translation in bicistronic and bicistronic splicing reporters. This also further demonstrated cap-independent translation of GGGGCC repeats. We now included this data in Fig. 1j,k, 6c and Supplementary Fig. 1c,3b.

Reviewer #2 (Remarks to the Author):

This is an improved manuscript that reports on the expression of polypeptides from the C9orf72 GGGGCC repeat containing mRNA. The main points established are that the mechanism for initiation seems to follow the IRES-mediated initiation pathway and that the relative level of expression is enhanced by eIF2 α phosphorylation. It is not clear if this upregulation is due to the loss of competition with the cap-dependent pathway which tends to be dominant in uninhibited cells or not. This reviewer anticipates that an acceptable manuscript can be generated without additional experimentation.

Response: We appreciate the reviewer's recognition of our improved manuscript. As suggested by reviewer 1, we included data of cap-dependent translation inhibition using mTOR inhibitors. There was no upregulation of cap-independent translation when the cap-dependent translation was profoundly inhibited. This may suggest the stress induced RAN translation elevation is not just due to the loss of competition.

Comments

1. page 6 – line 16 – The fact that translation occurs via an IRES element is NOT an indication that the mRNA is uncapped, but rather that the cap does not play a role in the translational efficiency.

Response: We agree with this point, and we think it has no conflict with what we described for the bicistronic reporter. For the bicistronic reporter, the RNA has 5'-cap and so does the first open reading frame (AUG-FLuc). But the translation of the second open reading frame (C9R-NLuc) does not need 5'-cap.

2. Page 10 – line 4 – The authors should make a clear distinction between the normal ATF4 mRNA and the splice variant that contains the IRES (ref. 55). Wild type ATF4 expression is

controlled via an upstream ORF while the splice variant is expressed via an IRES element, two very different mechanism but which seem to be run together in this paragraph.

Response: Thank you very much for raising this point. We now edited the writing to describe the two events more clearly.

3. The affects seen in Figure 6c seems rather small (50% at best) compared to what is seen in panel d (6 to 8-fold induction) and thus may not truly be reflecting the same phenomena.

Response: We think the chemical treatment is generally more potent than expression of an ectopic proteins. For example, every cell will be affected by a chemical, but not every cell can be transfected. The ectopic expression level of eIF2 α -S51D was very low compared to the endogenous eIF2 α (Supplementary Fig.2d), as we mentioned in the previous response. This could be one major reason that the chemical treatment gave rise to more changes. Nevertheless, the experiments of both phopho-eIF2 α inhibitor treatment and phospho-mimetic mutant expression showed that eIF2 α phosphorylation plays important roles in repeat translation elevation.

4. Figure 2 – Why are the results different depending on whether a mono- or dicistronic mRNA is tested (i.e. inhibition with the monocistronic, stimulation with the bicistronic)? Here and elsewhere, it would be good to also indicate what the absolute value is for the reference (or 1 value) in panel b (and other figures also).

Response: We appreciate this point. We think this data showed the cap-independent repeat translation is more potently enhanced by stress, compared to the cap-dependent translation. Taken consideration of both reviewers' opinion, we now separated the data of monocistronic reporter and bicistronic reporter in different graphs, as the goal of this experiment is to compare the translation level at stress conditions with the basal level of each individual reporter. The comparison of the relative basal expression level (without stress treatment) of all the three reporters (monocistronic, bicistronic, bicistronic splicing) is shown in Fig. 5d. We also have added the relative translation level from in vitro transcribed RNA with and without 5'-m7G cap (Fig. 1l). We think these provide more accurate information on how the basal level of cap-independent translation is compared to cap-dependent translation of GGGGCC repeats.

Minor comments

1. page 3, line 17 "...toxic dipeptides .." should be "toxic polydipeptides".

Response: Thank you! We now corrected the writing.

2. Page 6, line 6 – poly-A tail should be poly(A) tail

Response: Thank you! We now corrected the writing.

3. Page 7 – line 10 – "... and the cap binding protein ..." It is not clear as to whether this means eIF4F or the cap-recognition subunit of eIF4F which is eIF4E.

Response: Thank you for the suggestion. We now specified the description.

4. Page 8 – line 16 – While some IRES-containing mRNAs may be either not inhibited or upregulated with eIF2alpha phosphorylation, a number of mRNAs (GCN4, ATF4, etc.) are upregulated when ternary complexes are reduced and this regulation is affected by an upstream ORF(s).

Response: Thank you for the suggestion. We now modified our writing accordingly.

5. Page 13 – line 21 “..the spliced intronic RNA, which is not capped at the 5’ end.” Is there a reference for this or is this assumed with the idea that capping only occurs once with the native transcript? Secondly, in the absence of a cap and a poly(A) tail, it would normally be anticipated that the mRNA here would be short lived and perhaps not even exit the nucleus. Is there an explanation for why this intronic sequence appears in the cytoplasm (i.e. perhaps some unique feature of the repeat)?

Response: Thank you for the good points. We now added discussion about possible explanations how an intron appears in the cytoplasm. We didn’t find any literature reporting introns can be capped.